# Mechanical and Thermal Properties of Basalt Fibre Reinforced Polymer Lamellas for Renovation of Concrete Structures

**DOI:** 10.3390/polym14040790

**Published:** 2022-02-18

**Authors:** Szymon Grzesiak, Matthias Pahn, Andreas Klingler, Emmanuel Isaac Akpan, Milan Schultz-Cornelius, Bernd Wetzel

**Affiliations:** 1Department of Civil Engineering, Technical University of Kaiserslautern, 67663 Kaiserslautern, Germany; szymon.grzesiak@bauing.uni-kl.de (S.G.); milan.schultz-cornelius@bauing.uni-kl.de (M.S.-C.); 2Leibniz-Institut für Verbundwerkstoffe GmbH (IVW), Erwin-Schrödinger-Straße 58, 67663 Kaiserslautern, Germany; andreas.klingler@ivw.uni-kl.de (A.K.); emmanuel.akpan@ivw.uni-kl.de (E.I.A.); bernd.wetzel@ivw.uni-kl.de (B.W.)

**Keywords:** basalt fibre reinforced polymer, reinforced concrete, strengthening

## Abstract

The level of energy consumption in renovation activities of buildings has huge advantages over the demolition of old buildings and the construction of new structures. Such renovation activities are usually associated with the simultaneous strengthening of their elements, such as externally bonded carbon fibre reinforced polymer (CFRP) lamellas or sheets on vertical and horizontal surfaces as structural reinforcements. This means the process of refurbishing a building, as well as the raw materials themselves have a significant impact on CO_2_ emissions and energy consumption. This research paper demonstrates possibilities of replacing state of the art, highly energy-intensive CFRP lamellas with basalt fibre reinforced plastics as energy-efficient structural reinforcements for building constructions. The mechanical and thermal properties of basalt fibre reinforced polymer (BFRP) composites with variable matrix formulations are investigated. The article considers macro- and microstructures of innovative BFRP. The investigations focus on fibre–matrix interactions with different sizing formulations and their effect on the tensile strength, strain as well as modulus of elasticity.

## 1. Introduction

The renovation rate of existing buildings exceeds the construction of new buildings and has been increasing constantly in the last years [1]. The use of renovations as opposed to new constructions have recently been implemented in industrial, residential and commercial buildings [2]. This will likely increase in the future as the society gains awareness of the advantages of renovation over new constructions [3]. The most popular way to strengthen the tension zone in reinforced concrete (RC) elements is to use layers of concrete reinforced with textile- and fibre-reinforced polymers (FRP) like lamellae, sheet [4] or rebar [5]. These externally bonded (ED) reinforcements are already successfully applied to strengthen existing buildings. In concrete repair, the FRP retrofit or lamellae are usually attached to the surface of the concrete using reactive adhesives. Installation can also be done by placing the lamellae a small distance into the surface of the concrete [6]. Established for decades, reinforcing materials made of carbon-fibre-reinforced plastics have satisfying mechanical properties. However, carbon fibres have high amounts of embodied energy (total energy intensity of carbon fibre is estimated to be 284 MJ/kg [7,8,9]) and are generally expensive. On the other hand, basalt fibres are sustainable materials with comparatively lower amount of embodied energy (total energy intensity of basalt fibre of 18 MJ/kg [10,11,12]. Replacing carbon with basalt fibres in FRP for concrete renovation will increase environmental sustainability. The manufacturing cost of basalt fibres are also lower than carbon fibres. Their application in the manufacturing of lamella for strengthening of RC structures will have cost and environmental advantages over carbon fibre lamella [13]. A good understanding of the mechanical properties of BFRP lamellae will enable a cost-effective and sustainable design of new retrofitting techniques in building constructions.

A few studies have reported the mechanical properties of basalt-fibre-reinforced composites [14]. Some studies focused on the mechanical properties of BFRP bars [15,16]. Nonetheless, a few studies investigated the mechanical properties of BFRP laminates [17]. Chen et al. [18] examined the mechanical properties of BFRP under quasistatic and dynamic loading conditions. Tensile strength, modulus of elasticity and failure strain were 1642.2 MPa, 77.9 GPa and 0.021 respectively. Hashim et al. [19] examined the effect of matrix modification with silica particles on the tensile behaviour of BFRP composites under static conditions. Results showed a positive influence of silica particles on the BFRP mechanical properties. The particles increased the tensile strength of the composite product, for example, addition of 25 wt.% of silica particles increased the tensile strength from 400 MPa to 640 MPa. They enhanced the fibre–matrix interphase bonding. Azimpour-Shishevan et al. [20] investigated the effect of thermal cycling on properties of BFRP composites. Samples were subjected to temperature cycles (the cycle goes from room temperature to +120 °C, then from +120 °C to −40 °C and finally from −40 °C to room temperature) for a specified number of times and the mechanical properties were examined. Results showed that the tensile strength, modulus and inter laminar shear stress of the BFRP increased with increasing number of cycles until 80 cycles but decreased with further increase. The initial increase in properties was attributed to the effect of postcuring. Hu et al. [21] studied the shear modulus of BFRP laminates at different temperatures. Results showed that increase in temperature resulted to a degradation in mechanical performances due to degradation of the epoxy resin matrix. Lu and Xian [22] studied the combined effects of sustained tensile loading and elevated temperatures on the mechanical properties of a pultruded BFRP plates. It was observed that the temperatures affected tensile strength and modulus. For example, at 80 °C the tensile strength decreased by 9.8% and tensile modulus by 1.9%. The higher the exposure temperature the greater the resulting degradation. To improve the mechanical properties of BFRP composites, Matykiewicz et al. [23] studied the effect of different matrix modifications on the properties of basalt-fibre–epoxy composites. The BFRP were modified with zeolite or silsesquioxane particles and the mechanical properties determined under static conditions. The addition of particles decreased the tensile strength from 300 MPa to 270 MPa and the elastic modulus from 85 GPa to 66 GPa. Li et al. [24] examined the strain rate effects on tensile strength, Young’s modulus, and failure strain of BFRP under quasistatic and dynamic loadings. It was observed that the type of resin did not influence the tensile properties of BFRP composites, because the strength of epoxy resin is very small as compared to that of the fibre. 

The aim of this work is to investigate the effect of different matrix formulations and fibre sizing on the mechanical properties of basalt fibre reinforced polymer composites. Furthermore, the addition of core-shell rubber (CSR) particles has been tested in the matrix formulation. The CSR nanoparticles are expected to reduce or slow down the damage progression.

## 2. Methodology

### 2.1. Materials

The basalt fibre rovings in the study had a linear density of 2400 Tex (gram per kilometre), and an average diameter of 13 μm. The basalt fibres were obtained from Deutsche Basalt Fibre (DBF) GmbH, Frankfurt am Main, Germany. The basalt fibres were supplied with a single and double silane-based sized layer. As resins Biresin CR141 and CR144, as well as the hardener CH141 and the accelerator CA141 were obtained from Sika Deutschland GmbH [25]. CH141 and CA141 were used as curatives for CR141, Aradur CH917 (hardener) and DY070 (accelerator) were obtained from Huntsman Corporation and served as curatives for the second resin system. The curatives are chemically the same yet supplied by different companies. Core-shell rubber (Polybutadiene) nanoparticles (KaneAce MX153 [26]) were supplied by Kaneka Belgium N.V. as a master batch. The master batch contains 33 wt.% of core-shell rubber particles with a diameter of approximately 50–100 nm, being homogeneously dispersed in a diglycidyl ether of a bisphenol A-based epoxy resin system.

### 2.2. Manufacturing of BFRP-Lamellae

Basalt fibre lamella (BFL) were pultruded using the pultrusion line of the CG-TEC GmbH, Spalt, Germany (cf. Table 1). Two resin formulations were used for the pultrusion process: A—containing CR144, CH917 and DY070 and B—containing CR141, CH141 and CA141, each mixed in stoichiometric proportions. For some samples, the resins were modified with 5 wt.% core-shell rubber particles. The processability of MX153 was found to be challenging. Heating the modified resin system to higher temperatures to decrease the viscosity for processing was found to reduce its pot life.

In the pultrusion process, fibres were rolled out from the roving into the production line through a guided plate. The number of rovings for the selected batch was between 69 and 72. Each roving consisted of 6000 fibres. Due to the application of different sizing strategies, i.e., sized once vs. sized twice, the effective fibre diameter changed. The fibres then went through the resin impregnation bath at a speed of 6 mm/s. The impregnated fibres further passed through a preformer (cross-section 32 mm × 3 mm in Figure 1) to the curing chamber. The 1 m-long chamber was divided into three equal compartments operating at temperatures of 120 °C, 130 °C and 165 °C, respectively. The pulling force, which was transmitted to the BFRP-profile by a caterpillar pulling machine, was equal 2.2 kN.

### 2.3. Charaterization

#### 2.3.1. Fibre Volume Fraction

The fibre volume fraction of the pultruded lamella was determined via grey-scale analyses of light microscopy images [27]. Samples were embedded in an epoxy resin then grounded and polished using an automatic grinding machine. Grinding was performed with water with sequential change in emery papers from P400 to P4000 grit. The grounded samples were subsequently polished with a diamond solution of 3 and 1 µm. Fibre volume content and porosity of the samples were analysed using a Leica light microscope equipped with a grey scale analysis software.

#### 2.3.2. Tensile Test

Tensile tests were conducted of the basalt fibre lamella according to EN 2561 standard [28] using a testing machine (Schenck Hydropuls, Darmstadt, Germany) with a load cell of 250 kN [29]. Samples with a dimension of 250 mm × 15 mm × 3 mm [30] were conditioned in a humidity chamber at a temperature of 20 °C and a relative humidity of 60% for 48 h [31]. The samples were loaded in the fibre direction at a constant speed of 2 mm/min until failure (see Figure 2). Strain measurements were conducted using an extensometer, attached to the samples. To ensure repeatability, five specimens were tested for each sample. The tensile strength was calculated using Equation (1). Where: $P_{max}$ is the maximum load, dimensions *b* and *h* are the width and thickness of the samples, respectively. The features are measured with an electronic caliper at the beginning of the tests.
(1)$$\sigma_{t}=\frac{P_{max}}{b\cdot h}$$

Modulus of elasticity was calculated according to EN 2561 [28] using Equation (2). Where: $\varepsilon_{A}\;$and $\varepsilon_{B}$ are the strain parallel to the fibre direction corresponding to $0.1\cdot P_{max}$ and $0.5\cdot P_{max}$, respectively.
(2)$$E_{11}=\frac{0.4\cdot P_{max}}{b\cdot h\cdot \left[\varepsilon_{B}-\varepsilon_{A}\right]}$$

#### 2.3.3. Thermomechanical Analysis

The viscoelastic properties of the different lamellae were investigated by dynamic mechanical analysis (DMA, Q800, TA Instruments, Heidrun Klement, Germany) in a double cantilever setup at a frequency of f = 10 Hz. The rectangular specimens of 30 mm × 5 mm × and 2 mm were loaded perpendicular to the fibre direction. The temperature was varied from −100 °C up to 250 °C at a heating rate of 2 °C/min. The glass transition temperature was determined as the peak value of the mechanical damping (tan δ) [32].

#### 2.3.4. SEM Analysis

Structural analyses were performed using a scanning electron microscope (Zeiss Supra 40VP, Oberkochen, Germany) at various magnifications, deploying a secondary electron detector. The accelerating voltage was 5 kV. Surfaces to be examined were sputtered with a gold-palladium layer at *I* = 40 mA for 80 s (Balzers Sputter Coater SCD050, Balzers, Liechtenstein).

## 3. Results and Discussion

### 3.1. Fibre Volume Fraction

The fibre content has an impact not only on the cross-linking process of the resin system [33] but also, as a result, on mechanical properties, such as the elastic modulus and the tensile strength. Fiore et al. [34] summarised the influence of different fibre contents in BF composites on the various properties. As has been noted by Amuthakkannan et al. [35], the higher the BF content is, the higher is the tensile strength. Therefore, to quantify the fibre volume content of the BFRP lamella, corresponding samples are extracted from the lamella and embedded in a resin. The specimens have then been polished, and the fibre volume content was calculated using greyscale analysis based on light microscopy images (see Figure 3). Fibre volume fraction and porosity of the developed BFL are shown in Table 2.

The BFL show a fibre volume content of approx. 60 to 70 vol.%. The unmodified and double sized system (A2S) shows the highest fibre volume content. Modification with MX153 slightly reduces v_f_ (sample Amod1s). In the lamella with double sized fibres and core-shell rubber particles, there is a further reduction in the fibre volume content (Amod2s). 

Another parameter to be considered is the pore concentration in fibre reinforced composites. The BFL show a pore content of approx. 1.0 to 1.4 vol.%. Pores and inhomogeneities have an extremely detrimental effect on the stiffness of a material. Pores can be introduced into the materials during production process or reduced by the addition of inert or active filler materials [36]. The high impregnation quality leads to a ductile material, fracture-resistant and stiffness of matrix. The pore content negatively affects the durability of the manufactured BFL. Particular attention should be given to prevent bevor crack development in the discontinuity of the material’s microstructure in corrosive media.

### 3.2. Tensile Properties

Figure 4a depicts a representative stress-strain behaviour of a basalt fibre lamella. The figure shows that the lamella fails in a brittle manner, which is in line with a previous study on the static mechanical behaviour of woven basalt fibre reinforced epoxy composites [37,38]. As evidenced in Figure 4a the composites fail primarily by fibre breakage and debonding from the matrix. 

Figure 4b shows the variation of tensile strength and modulus of the developed basalt fibre lamella with respect to the applied sizing (1S and 2S, cf. Section 2.1) and matrix modification (mod). It is evident from the figure that the matrix modification led to a slight decrease in tensile strength. The elastic modulus shows almost no significant difference between the treatments. Composites with one-time sized fibre and no matrix modification show the highest tensile strength (1562 MPa) followed by the composite with double sizing and no matrix modification. 

For orientation, Figure 4c compares the tensile strength of commercial steel [39] and carbon fibre lamella with the newly developed basalt fibre lamellae. The tensile strain of BFL is in the range of 2.5% and 2.7%. The basalt fibre lamella surpasses steel but not the carbon fibre lamella in tensile performance. However, the strain at maximum load of BFL is superior to the strain at maximum load of CFRP based lamellae. 

Figure 4d illustrates the synergistic effect of the nanoparticle matrices and the fibre sizing on the tensile strength. BFLs with double sized fibres and matrix modification show lower tensile strengths, which can be attributed to a reduced fibre volume fraction v_f_. The fibre volume fraction decreases since the number of rovings in processing for the selected batch also decrease. This is due to the different fibre diameters with single and double sizing, which could be passed through a preformer to the curing chamber. Double sizing and matrix modification synergistically influence the difficulty of pultrusion of the BFL. On the other hand, composites with single sizing and nanoparticle modification show a moderate fibre volume fraction and consequently moderate tensile strength. It is postulated that the synergistic effect becomes lower when single sized fibres are used resulting in ease of processing and consequently an elevated fibre volume fraction is obtained. Composites without matrix modification show the highest fibre volume fraction and consequently higher tensile strength. Here, it is deduced that the driving force for the stated synergistic effect is the increase in viscosity of the resin as a result of addition of the nanoparticles (see Figure 5). However, this effect becomes stronger when the thickness of the sizing is increased.

### 3.3. Thermomechanical Properties

To investigate the influence of the modifiers and the applied fibre sizing on the thermomechanical properties of the cured BFRP lamellae, dynamic mechanical analyses were performed. Figure 6a shows the temperature dependent development of the mechanical damping coefficient tan δ of the various basalt fibre systems. All BFLs show a quite similar dynamic-mechanical behaviour, i.e., the damping increases slowly with increasing temperature up to the region of the dynamic glass transition, i.e., the transition from the solid to the viscoelastic state. This is also reflected in a sharp drop of the storage modulus (Figure 6b). The data show dynamic glass transition temperatures T_g_ in the range from 125.6 °C to about 135 °C, whereas the CR141 systems (B) show the highest values. The mechanical damping is strongly dependent on the present fibre volume content of the samples and varies in the range from 0.356 and 0.432. The basalt fibre has an approximately five times lower thermal conductivity (0.031–0.038 W/mK [40,41,42]) compared to the epoxy resin system (0.1 to 0.2 W/mK [43,44]), which can influence the heating process (heat flux) of the sample and thus the relaxation behaviour. The values are summarised in Table 3. There was no discernible postcuring of the systems, which would have been indicated by an increasing modulus after passing the region of the glass transition.

Conclusively, it seems as if neither the fibre sizing (strategy) nor the matrix modification have an effect on the thermomechanical behaviour of such lamella, at least in the investigated temperature range.

### 3.4. SEM Analyses

Figure 7 shows representative interlaminar fracture surfaces of the different basalt fibre reinforced lamella, obtained via scanning electron microscope (SEM) [45]. Qualitatively, there are only minor differences between the various fracture surfaces of the different material systems. However, the adhesion quality of the double-sized fibres seems visually better than the fibre–matrix interaction of the single-sized systems, especially Figure 7b,f show an enhanced fibre–matrix adhesion. The more sizing is present on the fibres the better the bonding quality to the matrix system. Even though, this enhanced interaction quality did not show a significant improvement of the tensile properties, as shown in Figure 4b, it is expected to yield a higher resistance to interlaminar crack propagation, i.e., enhanced fracture resistance of the lamella.

## 4. Conclusions

This paper presented a study on the mechanical properties of BFRP lamellae modified with core-shell rubber nanoparticles under the influence of a single and double sizing strategy. The goal was to investigate the influence of a matrix modification, while, simultaneously improving the fibre matrix interface properties, on the mechanical properties of basalt fibre reinforced polymer composites. The following conclusions can be drawn from this study:Basalt fibre reinforced polymer lamellae were successfully prepared in a pultrusion process.Tensile tests show the brittle character of BFRP composites. Quasistatic failure occurs immediately after leaving the linear-elastic region of the stress-strain curve.The type of fibre sizing strategy (sized once vs. sized twice) affects the effective fibre volume of the BFRP lamellae. This is because the cross-section of the lamella, defined by the tooling, is always the same and thus, the number of fibres pultrude through the cross-section needs to be reduced if the fibre diameter increases. Hence, if more silane sizing is used, the apparent fibre diameter increases. This effect is superimposed by an increased resin viscosity in the case of a CSR nanoparticle modified resin. Accordingly, the tensile strength and Young’s modulus decrease, once $v_{f}$ is reduced. The more sizing is applied on the basalt fibres, the higher the adhesion between fibre and matrix in the BFRP. This positive aspect has not been reflected in the mechanical properties of the composites because more sizing is related to the lower fibre volume fraction. Subsequently, this causes the mechanical properties to decrease.The type of epoxy resin influences the properties of BFRP lamellae. The sample with resin A, which had higher tensile strength than the sample with resin B, shows a similar trend in mechanical properties of the pultruded polymer. Furthermore, the addition of the core-shell rubber particles changes the properties of the matrix.Dynamic mechanical analyses showed that the resin modification does not affect the (thermo-)mechanical properties of the BFL. Furthermore, the sizing does not alter the dynamic glass transition.


Future investigations will focus on materials having the same fibre volume concentration, to better assess the effect of matrix and sizing modifications. The higher load capacity of strips with different diameter of fibre located in one pultruded cross-section is expected. Investigations on bond behaviour on the concrete for a whole retrofitting system with new basalt fibre strips are required.

Other studies should focus on the thermal durability [46] and corrosion resistance of BFRP with an example of the methodology procedure shown in [47,48]. The safety life cycle of BFRP strips should be examined in fatigue tests. Also of importance is the adhesion of BFRP lamellae to the concrete, which should be tested.

## Figures and Tables

**Figure 1 polymers-14-00790-f001:**
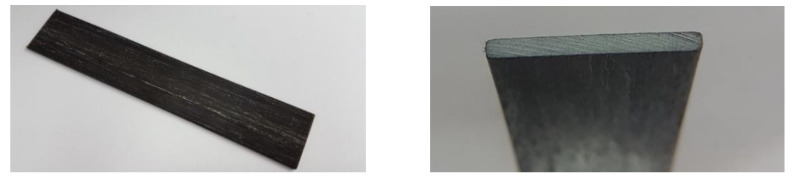
Profile pattern of BFL with cross-section 32 mm × 3 mm.

**Figure 2 polymers-14-00790-f002:**
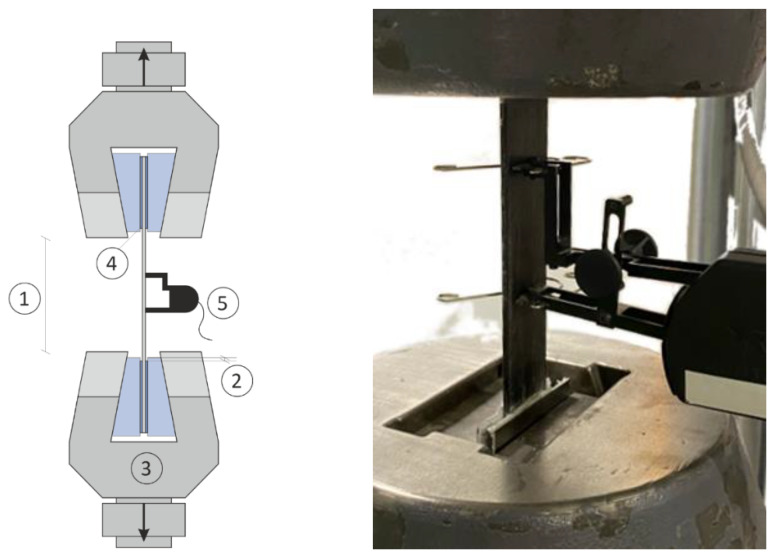
Schematic of tensile test: 1 = free length, 2 = projection of jaws, 3 = jaws, 4 = glued end tabs made from unidirectional glass-epoxy laminate, 5 = extensometer.

**Figure 3 polymers-14-00790-f003:**
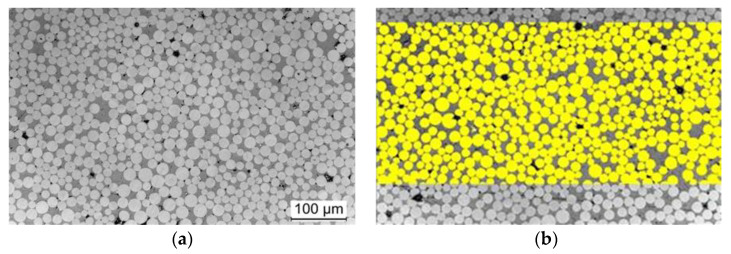
Determination of the fibre volume content of the developed BFL: (**a**) light microscopy image, (**b**) grey scale analysis.

**Figure 4 polymers-14-00790-f004:**
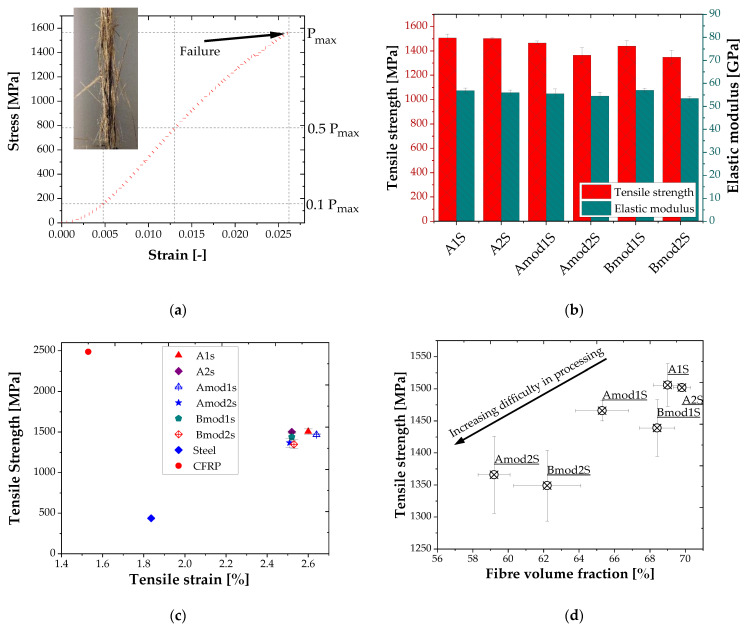
Tensile behaviour of the developed basalt fibre lamella: (**a**) typical stress strain curve of the BFL and fracture mode of the BFL, (**b**) effect of treatments on tensile strength and modulus of the BFL, (**c**) comparative tensile strength of BFL, CFRP and steel [39], (**d**) synergistic effect of nanoparticle modification and sizing on the tensile performance of the BFL.

**Figure 5 polymers-14-00790-f005:**
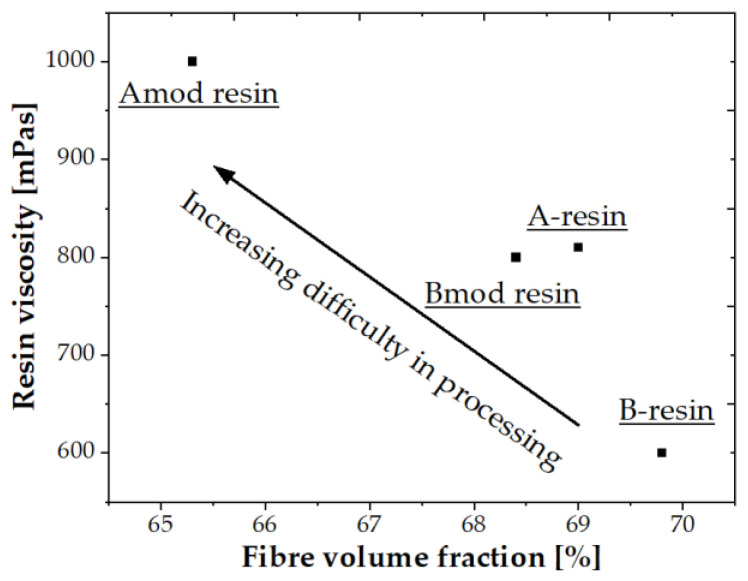
Correlating resin viscosity and fibre volume fraction of BFL.

**Figure 6 polymers-14-00790-f006:**
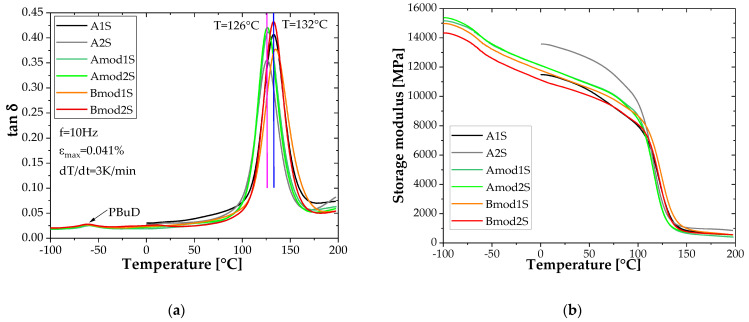
Temperature-dependent materials’ behaviour: (**a**) mechanical damping tan δ, (**b**) storage modulus.

**Figure 7 polymers-14-00790-f007:**
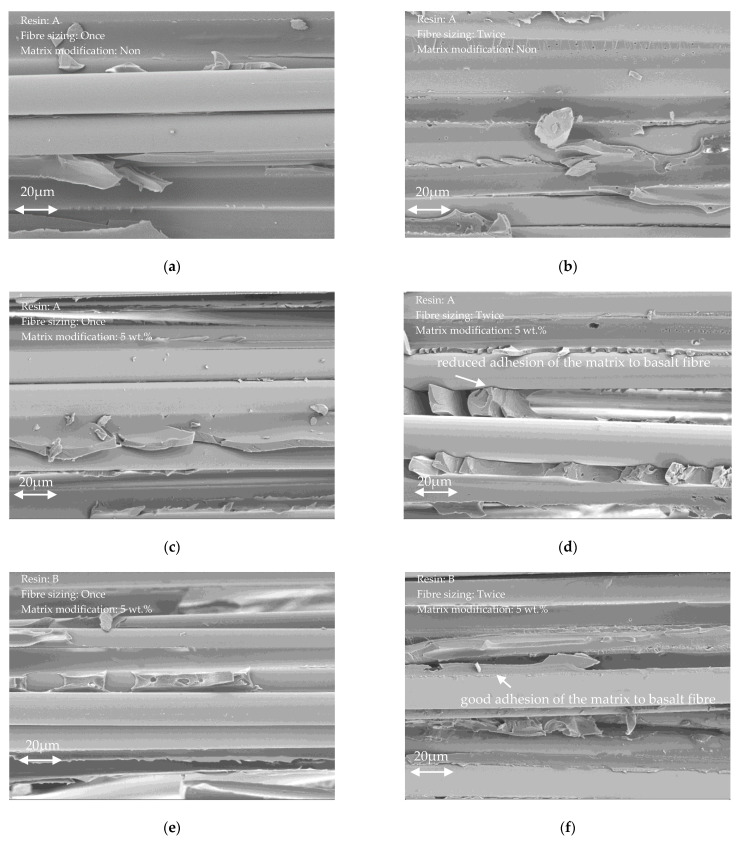
SEM analyses of the developed basalt fibre lamella: (**a**) A1S, (**b**) A2S, (**c**) Amod1s, (**d**) Amod2s, (**e**) Bmod1s, and (**f**) Bmod2s.

**Table 1 polymers-14-00790-t001:** Sample designations.

No	Description	Fibre Sizing	Matrix Modification	Resin Formulation
1	A1S	Once	Non	A
2	A2S	Twice	Non	A
3	Amod1S	Once	5 wt.%	A
4	Amod2S	Twice	5 wt.%	A
5	Bmod1S	Once	5 wt.%	B
6	Bmod2S	Twice	5 wt.%	B

**Table 2 polymers-14-00790-t002:** Fibre volume fraction and porosity of the developed BFL.

No	Description	Fibre Content (%)	Pore Content (%)
1	A1S	69.0 ± 0.8	1.30 ± 0.20
2	A2S	69.8 ± 0.5	0.98 ± 0.07
3	Amod1S	65.3 ± 1.5	1.40 ± 0.20
4	Amod2S	59.2 ± 0.9	1.20 ± 0.30
5	Bmod1S	68.4 ± 1.0	1.10 ± 0.10
6	Bmod2S	62.2 ± 1.9	1.20 ± 0.30

**Table 3 polymers-14-00790-t003:** Thermomechanical properties of the developed BFL.

No	Description	Glass Transition Temperature [°C]	Mechanical Damping Tan δ
1	A1S	132.8	0.407
2	A2S	125.6	0.356
3	Amod1S	126.8	0.412
4	Amod2S	126.1	0.421
5	Bmod1S	135.0	0.377
6	Bmod2S	132.8	0.432

## Data Availability

Not applicable.

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
