# Peer review of "Mechanical and Thermal Properties of Basalt Fibre Reinforced Polymer Lamellas for Renovation of Concrete Structures"

_polymers, 2022, doi:10.3390/polym14040790_

Round 1

Reviewer 1 Report

This is an interesting article about the important problem of restoring old buildings and the construction of new structures. It is extremely important to be able to increase the durability of new buildings as well as the possibility of restoring older structures. In addition, the use of modern solutions, i.e. the use of basalt fiber reinforced plastics as proposed in the article, allows for the reduction of CO2 emissions and energy consumption.

I believe that after some revision it would be interesting article for readers of Polymers. 

  1. All abbreviations should be explained the first time they are used in the text e.g. page 1, line 34 - RC; page 1, line 44 - basalt fibres (BR)
  2. Figure 5 is completely unreadable, please replace it with a higher resolution graph!
  3. The manuscript requires editorial corrections and linguistic verification (especially in the section "conclusions").

Reviewer 2 Report

The manuscript presents experimental results on the properties of BFRP sheets. two different matrix formulations were used in the composite. Also, the effect of the addition of core-shell rubber particles to the formulation has been investigated. The following comments are suggested to be considered.

  • Lns. 42-44: Is there any comparative study regarding the manufacturing energy of CFRP and BFRP? It should be cited and more details regarding the energy consumption should be given to support the statement of the authos.
  • Lns. 58-59: The properties are not given in a respective way, correction is required. 
  • Fig. 4a: The stress-strain curve is not clear, use bold lines.
  • Fig. 4c: The scale of the vertical axis should be modified to read the marks of steel and CFK properly.
  • The resolution of Fig. 5 is not good. One cannot read the text on the figure. It needs to be improved.
  • It is stated in the conclusion that fiber sizing enhanced the adhesion between the fiber and matrix. The sentence has to be completed to indicate why this has not been reflected on the mechanical properties of the composites.
  • There are some spelling and grammatical mistakes along the manuscript. Also, many sentences have to be rephrased. The manuscript has to be thoroughly reviewed and corrected.

Round 2

Reviewer 2 Report

The authors have addressed some of the comments offered by the reviewers but not all of them.

  • The curve in Fig. 4a is still not clear. The text on the figure "Rapture failure" has to be corrected to "Rupture failure".
  • The vertical scale in Fig. 4c should be modified to include the point of CFRP.
  • The English is still poor in some locations and needs to be corrected.
